# Clinical factors as prognostic variables among molecular subgroups of endometrial cancer

**Anne Kolehmainen**[1], **Annukka Pasanen**[2], **Taru Tuomi**[1], **Riitta Koivisto-Korander**[1], **Ralf Bützow**[1,2], **Mikko Loukovaara**[1] *

1 Department of Obstetrics and Gynecology, Helsinki University Hospital and University of Helsinki, Helsinki, Finland, 2 Department of Pathology, Helsinki University Hospital and Research Program in Applied Tumor Genomics, Faculty of Medicine, University of Helsinki, Helsinki, Finland

* mikko.loukovaara@hus.fi

## Abstract

### Background

Clinical factors may influence endometrial cancer survival outcomes. We examined the prognostic significance of age, body mass index (BMI), and type 2 diabetes among molecular subgroups of endometrial cancer.

### Methods

This was a single institution retrospective study of patients who underwent surgery for endometrial carcinoma between January 2007 and December 2012. Tumors were classified into four molecular subgroups by immunohistochemistry of mismatch repair (MMR) proteins and p53, and sequencing of polymerase-$\epsilon$ (*POLE*). Overall, cancer-related, and non-cancer-related mortality were estimated using univariable and multivariable survival analyses.

### Results

Age >65 years was associated with increased mortality rates in the whole cohort (n = 515) and in the "no specific molecular profile" (NSMP) (n = 218) and MMR deficient (MMR-D) (n = 191) subgroups during a median follow-up time of 81 months (range 1–136). However, hazard ratios for cancer-related mortality were non-significant for NSMP and MMR-D. Diabetes was associated with increased overall and non-cancer-related mortality in the whole cohort and MMR-D subgroup. Overweight/obesity had no effect on outcomes in the whole cohort, but was associated with decreased overall and cancer-related mortality in the NSMP subgroup, and increased overall and non-cancer-related mortality in the MMR-D subgroup. Overweight/obesity effect on cancer-related mortality in the NSMP subgroup remained unchanged after controlling for confounders. High-risk uterine factors were more common, and estrogen and progesterone receptor expression less common in NSMP subtype cancers of normal-weight patients compared with overweight/obese patients. No clinical factors were associated with outcomes in p53 aberrant (n = 69) and *POLE* mutant (n = 37) subgroups. No cancer-related deaths occurred in the *POLE* mutant subgroup.

**Funding:** This study was supported by Helsinki University Hospital research funds. The funder had

no role in study design, data collection and analysis, decision to publish, or preparation of the manuscript.

**Competing interests:** The authors have declared that no competing interests exist.

## Conclusions

The prognostic effects of age, BMI, and type 2 diabetes do not appear to be uniform for the molecular subgroups of endometrial cancer. Our data support further evaluation of BMI combined with genomics-based risk-assessment.

## Introduction

Endometrial cancer is the most common gynecologic malignancy in developed countries, with an incidence of 10 to 16 cases per 100,000 women in Western and Northern Europe [1]. Old age has a negative impact on the survival of endometrial cancer patients [2, 3]. Compared with younger patients, those >65 years have poorer overall survival and disease-specific survival whether or not they underwent lymphadenectomy, and irrespective of the presence of nodal metastasis [2]. Age ≥60 years is an independent predictor of locoregional relapses and disease-related death in stage I endometrial cancer [3]. In contrast, reports on the prognostic significance of body mass index and diabetes in endometrial cancer are inconsistent, as summarized in several meta-analyses [4–6]. The heterogeneous results may be explained by a number of factors, including differences in study design and selection of study subjects, methods of body mass index and diabetes assessment, lack of power, and choice of the outcome of interest. By merely assessing overall survival, the impact of potential risk factors on cancer-related survival may be unrecognized.

Earlier prognostic studies [2–6] were mostly conducted prior to the development of the molecular classification system for endometrial cancer [7]; therefore, they did not address the role of molecular subgroups in modifying the prognostic effect of clinical factors. The molecular classification system, identified through The Cancer Genome Atlas (TCGA), categorizes endometrial cancers into four distinct subgroups: polymerase-$\epsilon$ (*POLE*) ultramutated; microsatellite unstable hypermutated; copy-number low; and copy-number high. This categorization is based on: overall mutational burden; p53, *POLE*, and phosphatase and tensin homolog (*PTEN*) mutations; microsatellite instability; and histology [7]. Molecular subgroups are associated with different prognoses, so that: *POLE* ultramutated tumors have an excellent progression-free survival, microsatellite unstable hypermutated and copy-number low tumors an intermediate progression-free survival; and copy-number high tumors a poor progression-free survival [7].

Given the fact that the molecular subgroups can be considered to be different disease entities, each subgroup should ideally be investigated separately in clinical research studies. Based on the putative role of clinical factors in determining the prognosis of endometrial cancer, we wanted to examine the association of age, body mass index, and type 2 diabetes with patient outcomes among the different molecular subgroups.

## Materials and methods

### Study population and data collection

Patients underwent surgical treatment for stage I–IV endometrial carcinoma at the Department of Obstetrics and Gynecology, Helsinki University Hospital, between January 1, 2007 and December 31, 2012. Standard surgery included total hysterectomy and bilateral salpingo-oophorectomy. Lymphadenectomy was performed in selected patients. Adjuvant therapy was tailored according to stage and histologic findings at surgery. Patients with early-stage endometrioid carcinoma with high-risk features generally received either vaginal brachytherapy or

whole pelvic radiotherapy. Vaginal brachytherapy was mainly limited to patients in whom surgical nodal assessment was performed. Patients with non-endometrioid or advanced-stage endometrioid carcinoma received multimodality treatment with chemotherapy and radiation. Paclitaxel/carboplatin doublet was the standard chemotherapy regimen.

Clinicopathologic data were abstracted from institutional medical and pathology records. Weight and height were recorded at the time of endometrial cancer diagnosis. Overweight and obesity were specified as a body mass index 25–29.9 kg/m$^2$ and $\geq$30 kg/m$^2$, respectively, according to the World Health Organization (WHO) definitions. WHO class III obesity was defined as a body mass index $\geq$40 kg/m$^2$. Information on diabetes mellitus was captured by patient intake history at the time of initial consultation. Stage was determined according to the International Federation of Gynecology and Obstetrics guidelines revised in 2009 [8]. The cutoff for age was based on the finding that age >65 years is an independent poor prognostic factor in endometrial cancer [2]. The choice of 5 cm as a determinant for the analysis of tumor size was based on earlier literature according to which size approximating the entire uterine cavity is strongly associated with survival in stage I endometrial cancer [9]. Lymphovascular space invasion was defined as the presence of adenocarcinoma, of any extent, in endothelium-lined channels of uterine specimens outside the tumor.

Cause of death was mainly based on medical records. Missing data were complemented from death certificates. The study was approved by the local institutional review board and the National Supervisory Authority for Welfare and Health. Informed consent was waived because of the retrospective nature of the study.

## Molecular classification

Tumors were categorized into molecular subgroups according to the TransPORTEC classifier that recapitulates the four molecular subgroups of the TCGA as follows: "no specific molecular profile" (NSMP, surrogate to copy-number low in the TCGA classification system); mismatch repair deficient (MMR-D, surrogate to microsatellite unstable hypermutated); p53 abnormal (p53 abn, surrogate to copy-number high); and *POLE* mutant [10]. Minor adjustments were introduced to the TransPORTEC protocol. First, while our p53 and microsatellite instability analyses were solely based on immunohistochemistry, the TransPORTEC classifier uses a combination of *TP53* mutational testing and p53 immunohistochemistry to determine p53 status, and primarily the Promega microsatellite instability analysis for determination of microsatellite instability status. For tumors exhibiting low levels of instability, or from which extracted DNA quality is poor, immunohistochemistry of MMR proteins is performed. Second, the TransPORTEC classifier detects *POLE* exonuclease domain hotspot mutations by Sanger sequencing of exons 9 and 13, whereas we performed sequencing of exons 9, 13, and 14. Lastly, we did not exclude cases with multiple classifying alterations. Instead, they were categorized according to the alteration that determines the clinical outcome [11, 12].

Patients with adequate tumor samples for a tissue microarray were eligible for the study. For the construction of tissue microarray, histologic slides were reviewed by a pathologist and representative areas of each tumor were marked on the slides. Four duplicate 0.8 mm cores were drawn from the corresponding area of the paraffin blocks and a tissue microarray block was prepared. The following monoclonal antibodies were used for chromogenic immunohistochemistry: MLH1 (ES05, Dako, Santa Clara, CA, USA); MSH2 (G219-1129, BD Biosciences, San Jose, CA, USA); MSH6 (EPR3945, Abcam, Cambridge, UK); PMS2 (EPR3947, Epitomics, Burlingame, CA, USA); and p53 (DO-7, Dako). Tissue microarray slides were scanned with three-dimensional Histech Pannoramic 250 Flash II scanner by Fimmic Oy (Helsinki, Finland). Slide images were managed and analyzed with WebMicroscope Software (Fimmic Oy).

Virtual slides were scored by a pathologist blinded to clinical data. A second investigator examined equivocal cases and a consensus was reached. MMR protein status was considered deficient when we observed a complete loss of nuclear expression in carcinoma cells of one or more MMR proteins (MLH1, MSH2, MSH6, PMS2) detected by immunohistochemistry. Aberrant p53 staining was defined as strong and diffuse nuclear staining or completely negative ('null') staining in carcinoma cells. Weak and heterogeneous staining was classified as wild-type expression. Stromal cells and inflammatory cells served as internal controls for MMR and p53 stainings. Samples with scarce carcinoma cells or completely negative staining of the internal controls, when applicable, were discarded.

For DNA extraction, representative areas of formalin-fixed paraffin-embedded tumor tissue were macrodissected as identified by pathologist assessment. DNA was extracted by proteinase K/phenol-chloroform method. *POLE* exonuclease domain mutation (EDM) screening of hot spots in exon 9 (c.857C>G, p.P286R; c.890C>T, p.S297F), exon 13 (c.1231G>C, p.V411L), and exon 14 (c.1366G>C, p.A456P) was performed by direct sequencing. The following primers were used: Ex 9F (5'–3'): CCTAATGGGGAGTTTAGAGCTT; Ex 9R (5'–3'): CCCATCCCAG GAGCTTACTT; Ex 13F (5'–3'): TCTGTTCTCATTCTCCTTCCAG; Ex 13R (5'–3'): CGGGAT GTGGCTTACGTG; Ex 14F (5'–3'): TGACCCTGGGCTCTTGATTT; Ex 14R (5'–3'): ACAGGACA GATAATGCTCACC. Polymerase chain reaction products were sequenced on an ABI3730XI Automatic DNA Sequencer (Applied Biosystems, Foster City, CA, USA). Sequence graphs were analyzed both manually and with Mutation Surveyor (Softgenetics, State College, PA, USA). Only cases with good-quality sequence for all the examined four *POLE* hot spots were included in the analysis.

For further characterization, the following monoclonal antibodies were used for chromogenic immunohistochemistry on multicore tissue microarray slides: estrogen receptor-α (SP1, Roche/Ventana, Oro Valley, AZ, USA); progesterone receptor (clone 16, Novocastra, Newcastle upon Tyne, UK); and L1 cell adhesion molecule (L1CAM, clone 14.10, Covance, Princeton, NJ, USA). The cut-off for positive estrogen and progesterone receptor expression was set at 10% based on endometrial cancer and breast cancer studies [13–15]. For L1CAM expression, ≥10% of membranous staining was considered positive [16–18].

### Statistical methods

Chi-squared test was used for comparison of categorical variables, and analysis of variance and Kruskal-Wallis test for comparison of continuous variables after testing for normality by Shapiro-Wilk test. Hazard ratios for overall mortality, cancer-related mortality, and non-cancer-related mortality were estimated using univariable and multivariable Cox regression analyses. Cancer-related mortality was the main outcome measure. Variables with proven prognostic significance were entered as covariates in the multivariable model. These included stage [19], features of the primary tumor [20, 21], estrogen and progesterone receptor expression [13, 14, 22], L1CAM expression [16–18, 23], and adjuvant therapy [24]. Survival times were estimated using the Kaplan-Meier method. Differences between groups were compared using the log rank test. Survivals were calculated as the times from surgery to death. Statistical significance was set at $P < 0.05$. Data were analyzed using the Statistical Package for the Social Sciences version 25 software (IBM Corp., Armonk, NY, USA).

### Results

A total of 515 endometrial carcinomas were classified into molecular subgroups. Twenty cases (3.9%) displayed multiple molecular features. Three cases displayed *POLE* EDM and either MMR-D or p53 abn, and one case had all three molecular alterations. These were classified as

*POLE* EDM tumors [12]. Sixteen cases, classified as MMR-D tumors [11], displayed both MMR-D and p53 abn. The basic characteristics of the study population are summarized in Table 1. We found that *POLE* EDM is associated with younger age and lower body mass index, whereas p53 abn is associated with older age. The prevalence of type 2 diabetes was similar between molecular subgroups.

Median follow-up time was 81 months (range 1–136). During follow-up a total of 160 patients died, including 97 deaths related to endometrial cancer (Table 2).

Univariable Cox regression analyses were performed separately for overall mortality, cancer-related mortality, and non-cancer-related mortality (Table 2). Associations of clinical

**Table 1. Characteristics of the study population according to molecular subgroups.**

| | NSMP (n = 218) | *POLE* EDM (n = 37) | MMR-D (n = 191) | p53 abn (n = 69) | P |
|---|---|---|---|---|---|
| Age (years) [median (interquartile range)] | 66 (60–73) | 59 (53–68) | 70 (61–77) | 72 (66–78) | <0.0005 |
| Age >65 years | 116 (53.2%) | 11 (29.7%) | 121 (63.4%) | 52 (75.4%) | <0.0005 |
| Body mass index (kg/m$^2$) [median (interquartile range)] | 28.5 (24.3–33.2) | 25.1 (23.0–28.3) | 27.1 (23.3–32.7) | 27.3 (24.4–30.5) | 0.023 |
| Overweight/obese | 157 (72.0%) | 21 (56.8%) | 118 (61.8%) | 45 (65.2%) | 0.091 |
| World Health Organization class III obesity | 14 (6.4%) | 1 (2.7%) | 7 (3.7%) | 3 (4.3%) | 0.541 |
| Type 2 diabetes | 40 (18.3%) | 4 (10.8%) | 37 (19.4%) | 13 (18.8%) | 0.671 |
| Pelvic lymphadenectomy | 129 (59.2%) | 23 (62.2%) | 106 (55.5%) | 32 (46.4%) | 0.255 |
| Pelvic-aortic lymphadenectomy | 19 (8.7%) | 5 (13.5%) | 34 (17.8%) | 22 (31.9%) | <0.0005 |
| Stage | | | | | <0.0005 |
| IA | 123 (56.4%) | 28 (75.7%) | 84 (44.0%) | 22 (31.9%) | |
| IB | 42 (19.3%) | 6 (16.2%) | 44 (23.0%) | 18 (26.1%) | |
| II | 23 (10.6%) | 2 (5.4%) | 19 (9.9%) | 1 (1.4%) | |
| IIIA | 9 (4.1%) | 1 (2.7%) | 13 (6.8%) | 5 (7.2%) | |
| IIIB | 1 (0.5%) | 0 (0%) | 2 (1.0%) | 1 (1.4%) | |
| IIIC1 | 13 (6.0%) | 0 (0%) | 18 (9.4%) | 3 (4.3%) | |
| IIIC2 | 1 (0.5%) | 0 (0%) | 7 (3.7%) | 9 (13.0%) | |
| IVA | 0 (0%) | 0 (0%) | 0 (0%) | 0 (0%) | |
| IVB | 6 (2.8%) | 0 (0%) | 4 (2.1%) | 10 (14.5%) | |
| Histology | | | | | <0.0005 |
| Endometrioid carcinoma | 206 (94.5%) | 34 (91.9%) | 174 (91.1%) | 36 (52.2%) | |
| Clear cell carcinoma | 5 (2.3%) | 2 (5.4%) | 5 (2.6%) | 13 (18.8%) | |
| Serous carcinoma | 2 (0.9%) | 1 (2.7%) | 3 (1.6%) | 11 (15.9%) | |
| Carcinosarcoma | 2 (0.9%) | 0 (0%) | 3 (1.6%) | 7 (10.1%) | |
| Undifferentiated carcinoma | 3 (1.4%) | 0 (0%) | 6 (3.1%) | 2 (2.9%) | |
| Grade (For endometrioid only; n = 450) | | | | | <0.0005 |
| 1 | 141 (68.4%) | 21 (61.8%) | 79 (45.4%) | 5 (13.9%) | |
| 2 | 52 (25.2%) | 8 (23.5%) | 54 (31.0%) | 15 (41.7%) | |
| 3 | 13 (6.3%) | 5 (14.7%) | 41 (23.6%) | 16 (44.4%) | |
| Adjuvant therapy | | | | | <0.0005 |
| None | 33 (15.1%) | 6 (16.2%) | 20 (10.5%) | 7 (10.1%) | |
| Vaginal brachytherapy | 116 (53.2%) | 22 (59.5%) | 83 (43.5%) | 18 (26.1%) | |
| Pelvic radiotherapy | 28 (12.8%) | 6 (16.2%) | 35 (18.3%) | 12 (17.4%) | |
| Chemotherapy | 7 (3.2%) | 0 (0%) | 8 (4.2%) | 7 (10.1%) | |
| Chemotherapy and vaginal brachytherapy | 10 (4.6%) | 0 (0%) | 10 (5.2%) | 11 (15.9%) | |
| Chemotherapy and pelvic radiotherapy | 24 (11.0%) | 3 (8.1%) | 35 (18.3%) | 14 (20.3%) | |

Abbreviations: MMR-D, mismatch repair deficient; NSMP, no specific molecular profile; *POLE* EDM, polymerase-$\epsilon$ exonuclease domain mutation; p53 abn, p53 abnormal.

**Table 2. Univariable Cox regression analyses of overall, cancer-related and non-cancer-related mortality.**

| | Mortality | All | NSMP | *POLE* EDM | MMR-D | p53 abn |
|---|---|---|---|---|---|---|
| | | (n = 515) | (n = 218) | (n = 37) | (n = 191) | (n = 69) |
| | | HR (95% CI) | HR (95% CI) | HR (95% CI) | HR (95% CI) | HR (95% CI) |
| Age >65 Years | Overall † | 2.7 (1.9–3.9) P < 0.0005 | 2.8 (1.4–5.5) P = 0.004 | 4.6 (0.42–51) P = 0.212 | 2.6 (1.5–4.4) P = 0.001 | 0.95 (0.45–2.0) P = 0.900 |
| | Cancer-related ‡ | 1.7 (1.1–2.6) P = 0.016 | 2.0 (0.87–4.7) P = 0.102 | – | 1.3 (0.68–2.4) P = 0.461 | 0.74 (0.33–1.7) P = 0.471 |
| | Non-cancer-related § | 8.2 (3.5–19) P < 0.0005 | 4.8 (1.4–17) P = 0.013 | 4.6 (0.42–51) P = 0.212 | 23 (3.1–165) P = 0.002 | 2.7 (0.34–21) P = 0.353 |
| Overweight/ Obese | Overall † | 0.82 (0.60–1.1) P = 0.218 | 0.37 (0.21–0.68) P = 0.001 | 0.37 (0.033–4.0) P = 0.410 | 1.8 (1.1–3.0) P = 0.018 | 0.65 (0.34–1.2) P = 0.190 |
| | Cancer-related ‡ | 0.72 (0.48–1.1) P = 0.104 | 0.32 (0.15–0.71) P = 0.005 | – | 1.3 (0.72–2.5) P = 0.356 | 0.69 (0.33–1.5) P = 0.327 |
| | Non-cancer-related § | 1.0 (0.60–1.7) P = 0.954 | 0.46 (0.18–1.2) P = 0.101 | 0.37 (0.033–4.0 P = 0.410 | 3.1 (1.3–7.5) P = 0.013 | 0.56 (0.16–1.9) P = 0.364 |
| Type 2 diabetes | Overall † | 1.5 (1.1–2.2) P = 0.019 | 0.87 (0.39–1.9) P = 0.729 | 5.0 (0.45–55) P = 0.191 | 2.5 (1.5–4.0) P < 0.0005 | 0.95 (0.42–2.2) P = 0.907 |
| | Cancer-related ‡ | 1.0 (0.62–1.8) P = 0.876 | 0.61 (0.18–2.0) P = 0.421 | – | 1.8 (0.91–3.6) P = 0.089 | 0.51 (0.15–1.7) P = 0.266 |
| | Non-cancer-related § | 2.5 (1.5–4.3) P < 0.0005 | 1.3 (0.42–3.9) P = 0.674 | 5.0 (0.45–55) P = 0.191 | 3.6 (1.8–7.4) P < 0.0005 | 2.8 (0.79–10) P = 0.111 |

† N deaths = 160 (n = 43, n = 3, n = 76 and n = 38 for NSMP, *POLE* EDM, MMR-D and p53 abn, respectively)

‡ N deaths = 97 (n = 25, n = 0, n = 44 and n = 28 for NSMP, *POLE* EDM, MMR-D and p53 abn, respectively).

§ N deaths = 63 (n = 18, n = 3, n = 32 and n = 10 for NSMP, *POLE* EDM, MMR-D and p53 abn, respectively).

Abbreviations: CI, confidence interval; HR, hazard ratio; MMR-D, mismatch repair deficient; NSMP, no specific molecular profile; *POLE* EDM, polymerase-$\epsilon$ exonuclease domain mutation; p53 abn, p53 abnormal.

factors with the outcomes were first examined in the whole cohort. Old age (>65 years) was invariably associated with poor outcomes. Overweight/obesity (body mass index $\geq$25 kg/m$^2$) was not associated with any of the outcomes, whereas type 2 diabetes was associated with increased overall mortality and non-cancer-related mortality.

Associations were then examined separately for each molecular subgroup (Table 2). Old age was associated with increased overall mortality and non-cancer-related mortality in NSMP and MMR-D subgroups. Overweight/obesity and type 2 diabetes were associated with increased overall mortality and non-cancer-related mortality in the MMR-D subgroup. In the NSMP subgroup, overweight/obesity was associated with decreased overall mortality and cancer-related mortality, our main outcome of interest. For *POLE* EDM and p53 abn, significant associations between clinical factors and outcomes were not observed. Hazard ratios for cancer-related mortality were not calculable in the *POLE* EDM subgroup because there were no cancer-related deaths in this subgroup of patients.

Kaplan-Meier disease-specific survival analyses were performed separately for three body mass index categories in the NSMP subgroup. Disease-specific survival was similarly improved for overweight patients (body mass index 25–29.9 kg/m$^2$) and obese patients (body mass index $\geq$30 kg/m$^2$) compared with normal-weight patients (body mass index <25 kg/m$^2$) (Fig 1). Body mass index 25 kg/m$^2$ was therefore selected as the cut-off for further analyses.

Table 3 shows the proportions of various prognostic variables in normal-weight and overweight/obese patients with NSMP subtype cancer. High-risk histology, deep myometrial invasion ($\geq$50%), and lymphovascular space invasion were more common in normal-weight compared with overweight/obese patients. Moreover, estrogen and progesterone receptor

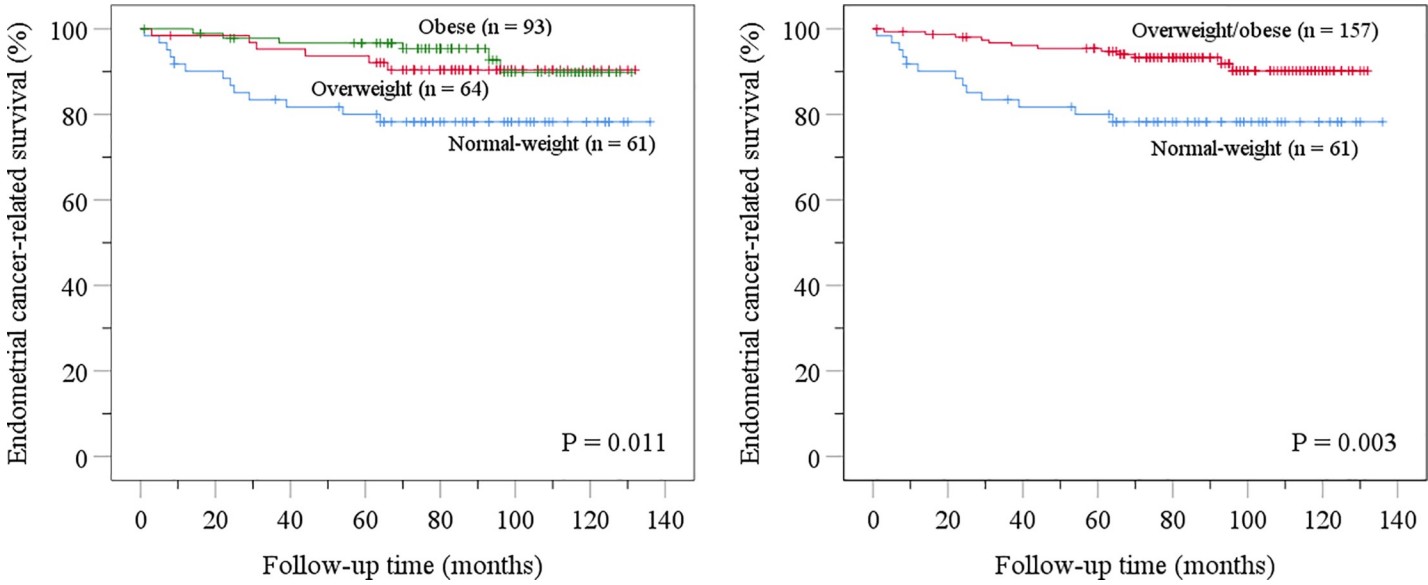

**Fig 1. Kaplan-Meier disease-specific survival analyses concerning body mass index in the "no specific molecular profile" subgroup.**

expression was less common in normal-weight patients. L1CAM expression and proportions of old patients (>65 years), stage II–IV cancers, large tumors (>5 cm), and adjuvant therapies received were not significantly different for normal-weight and overweight/obese patients.

**Table 3. Proportions of various prognostic variables in normal-weight and overweight/obese patients with "no specific molecular profile" subtype endometrial cancer.**

| | Normal-weight (n = 61) | Overweight/obese (n = 157) | P |
|---|---|---|---|
| Age >65 years | 33 (54.1%) | 83 (52.9%) | 0.870 |
| Stage II–IV | 18 (29.5%) | 35 (22.3%) | 0.265 |
| Histology | | | <0.0005 |
| Endometrioid grade 1–2 carcinoma | 44 (72.1%) | 149 (94.9%) | |
| Endometrioid grade 3 carcinoma | 9 (14.8%) | 4 (2.5%) | |
| Non-endometrioid carcinoma | 8 (13.1%) | 4 (2.5%) | |
| Myometrial invasion ≥50% | 30 (49.2%) | 53 (33.8%) | 0.035 |
| Tumor size >5 cm † | 17 (28.8%) | 27 (18.9%) | 0.120 |
| Lymphovascular space invasion | 21 (34.4%) | 28 (17.8%) | 0.008 |
| Estrogen receptor expression ‡ | 48 (80.0%) | 143 (94.1%) | 0.002 |
| Progesterone receptor expression § | 40 (66.7%) | 139 (89.1%) | <0.0005 |
| L1 cell adhesion molecule expression ⌐ | 6 (10.3%) | 8 (5.3%) | 0.187 |
| Adjuvant therapy | | | 0.537 |
| None or vaginal brachytherapy | 39 (63.9%) | 110 (70.1%) | |
| Pelvic radiotherapy | 7 (11.5%) | 21 (13.4%) | |
| Chemotherapy | 7 (11.5%) | 10 (6.4%) | |
| Chemotherapy and pelvic radiotherapy | 8 (13.1%) | 16 (10.2%) | |

† Data missing for 2 normal-weight and 14 overweight/obese patients

‡ data missing for 1 normal-weight and 5 overweight/obese patients

§ data missing for 1 normal-weight and 1 overweight/obese patient

⌐ data missing for 3 normal-weight and 5 overweight/obese patients.

**Table 4. Univariable and multivariable Cox regression analyses of cancer-related mortality for the "no specific molecular profile" subgroup.**

| | Univariable (n = 218) | | | Multivariable (n = 186) | |
| --- | --- | --- | --- | --- | --- |
| | N deaths = 25 | | | N deaths = 20 | |
| | N (%) | HR (95% CI) | P | HR (95% CI) | P |
| Overweight/obese | 157 (72.0%) | 0.32 (0.15–0.71) | 0.005 | 0.32 (0.11–0.92) | 0.034 |
| Stage II-IV | 53 (24.3%) | 5.3 (2.4–12) | <0.0005 | 6.1 (0.97–39) | 0.053 |
| Histology | | | <0.0005 | | 0.425 |
| Endometrioid grade 1–2 carcinoma | 193 (88.5%) | 1 | | 1 | |
| Endometrioid grade 3 carcinoma | 13 (6.0%) | 14 (5.6–33) | <0.0005 | 3.2 (0.49–20) | 0.228 |
| Non-endometrioid carcinoma | 12 (5.5%) | 6.6 (2.2–20) | 0.001 | 1.4 (0.19–10) | 0.736 |
| Myometrial invasion ≥50% | 83 (38.1%) | 5.7 (2.3–14) | <0.0005 | 2.9 (0.68–12) | 0.153 |
| Tumor size >5 cm † | 44 (21.8%) | 3.8 (1.7–8.4) | 0.001 | 1.4 (0.38–4.8) | 0.634 |
| Lymphovascular space invasion | 49 (22.5%) | 5.9 (2.7–13) | <0.0005 | 1.8 (0.58–5.4) | 0.315 |
| Estrogen receptor expression ‡ | 191 (90.1%) | 0.14 (0.063–0.32) | <0.0005 | 0.62 (0.11–3.5) | 0.587 |
| Progesterone receptor expression § | 179 (82.9%) | 0.28 (0.12–0.64) | 0.003 | 1.6 (0.42–6.3) | 0.483 |
| L1 cell adhesion molecule expression ⸓ | 14 (6.7%) | 7.1 (2.8–18) | <0.0005 | 3.0 (0.61–14) | 0.177 |
| Adjuvant therapy | | | 0.012 | | 0.273 |
| None or vaginal brachytherapy | 149 (68.3%) | 1 | | 1 | |
| Pelvic radiotherapy | 28 (12.8%) | 1.4 (0.40–5.2) | 0.573 | 0.20 (0.028–1.4) | 0.111 |
| Chemotherapy | 17 (7.8%) | 3.8 (1.2–12) | 0.022 | 0.78 (0.14–4.3) | 0.772 |
| Chemotherapy and pelvic radiotherapy | 24 (11.0%) | 4.2 (1.6–11) | 0.003 | 0.21 (0.026–1.7) | 0.141 |

† Data missing for 16 patients

‡ data missing for 6 patients

§ data missing for 2 patients

⸓ data missing for 8 patients.

Abbreviations: CI, confidence interval; HR, hazard ratio.

To assess the contribution of various clinicopathologic variables to patient outcome in the NSMP subgroup, we performed univariable and multivariable Cox regression analyses of cancer-related mortality (Table 4). All of the tested variables, i.e. overweight/obesity, stage, features of the primary tumor, estrogen and progesterone receptor expression, L1CAM expression, and adjuvant therapy, were associated with cancer-related mortality in unadjusted analyses. In the multivariable model, only overweight/obesity was found to have a significant independent effect on the outcome.

## Discussion

We explored the variation and prognostic significance of age, overweight/obesity, and type 2 diabetes in 515 women with endometrial carcinoma that were classified into molecular subgroups by immunohistochemistry of MMR proteins and p53, as well as *POLE* sequencing. Old age, overweight/obesity, and type 2 diabetes in the MMR-D subgroup, and old age in the NSMP subgroup were associated with increased overall mortality and non-cancer-related mortality. Overweight/obesity was associated with decreased overall mortality and cancer-related mortality in the NSMP subgroup. *POLE* EDM was associated with younger age and lower body mass index, whereas p53 abn was associated with older age, in accordance with previous studies [25–27]. For these subgroups, significant associations between clinical factors and outcomes were not observed. Hazard ratios for cancer-related mortality were not calculable in the *POLE* EDM subgroup because there were no cancer-related events in this subgroup of patients. Clinicopathologic characteristics generally varied among subgroups, which can be

explained by the fact that the molecular classifier employed in this study correlates with traditional prognostic factors of endometrial cancer [10].

To better understand the effect of body mass index on cancer-related mortality in the NSMP subgroup, we compared the proportions of various risk factors in normal-weight and overweight/obese patients in this subgroup, and found that high-risk uterine factors were more common in normal-weight patients. The presence of uterine risk factors is associated with increased risk for recurrences and poor survival, even in the absence of metastatic nodal disease [20, 21]. Lean patients in the NSMP subgroup also had a high-risk expression profile of molecular biomarkers. Expression of estrogen and progesterone receptors, known to be associated with improved endometrial cancer-specific survival [13, 14, 22], was less common in normal-weight patients compared with overweight/obese patients. Moreover, although not statistically significant, the expression of L1CAM, a poor prognostic factor in endometrial cancer [16–18, 23], was twice as common in normal-weight patients as in overweight/obese patients. When the prognostic effects of uterine risk factors and molecular biomarkers, along with overweight/obesity, stage and adjuvant therapy, were assessed in a multivariable model, only overweight/obesity had an independent effect on cancer-related mortality. This emphasizes the prognostic strength of body mass index in the NSMP subgroup, relative to other established risk factors.

This study is strengthened by the large sample size that allowed us to undertake analyses stratified into the various molecular subgroups of endometrial cancer. Admittedly, however, the smaller sizes of *POLE* EDM and p53 abn molecular subgroups may have precluded significant findings of clinical factors on outcomes in these subgroups. Information on cause of death was available for all of the deceased which allowed us to make a distinction in assessing not only overall mortality, but also cancer-related mortality, which can be considered the ideal outcome of interest when looking for causalities in cancer research. Knowledge of cancer-related deaths is especially important in endometrial cancer, for which competing causes of death are common; in the current study, 39% of deaths were secondary to causes other than endometrial cancer. Unlike most earlier studies, we were also able to distinguish between type 1 and type 2 diabetes. This improved the assessment of the true relation between diabetes and outcomes, as only type 2 diabetes appears to be prognostic in endometrial cancer [28].

Although our findings are limited to a single institution, the stage distribution and proportion of non-endometrioid carcinomas were comparable to an unselected cohort of 5,866 patients in the Gynecologic Oncology Group 210 surgical pathological staging study, in which the vast majority of tumors were early-stage endometrioid carcinomas [29]. The frequency of diabetes was captured by self-report, which could potentially lead to some misclassification. However, patient administered questionnaires, also used in our hospital, appear to be a reliable source of information in diabetes [30]. Information on the treatment of diabetes was unavailable for our study, which may partly distort the results, as metformin has been shown to improve overall survival and progression-free survival in patients with endometrial cancer [31].

Based on the current findings, endometrial cancer subgroups described by the TCGA represent not only unique genomic subgroups, but also entities that have distinct clinical prognostic features. Presuming that the proportions of molecular subgroups can vary across different study cohorts, the earlier contradictory findings [4–6] on clinical factors as prognosticators may be somewhat explained by the neutral effect of clinical factors on outcomes of *POLE* EDM and p53 abn subgroups, and partly opposing effect on outcomes of NSMP and MMR-D subgroups. Our findings may aid in patient counseling on the interrelation between dietary intake and prognosis, in addition to lending support to the implementation of body mass index in genomics-based risk-assessment. Future studies will probably show if refinement of

genomics-based characterization can explain the different prognosis of patients with identical molecular subgroups but unlike clinical characteristics. For example, inactivation of *PTEN*, whose mutational frequency differs between TCGA subgroups [7], has been suggested to determine the outcome of endometrial cancer patients in the context of body mass index [32, 33].

In summary, our findings suggest that the prognostic effects of old age, overweight/obesity, and type 2 diabetes are not uniform for the molecular subgroups of endometrial cancer. Thus, clinical factors should be assessed as prognostic variables in conjunction with the molecular subgroup. The present data suggest that the metabolic consequences of adiposity play different roles in the aggressiveness of endometrial cancer, depending on the molecular subtype. Further work is needed to identify molecular pathways and prognostic biomarkers specific to women with different clinical characteristics.

## Author Contributions

**Conceptualization:** Annukka Pasanen, Ralf Bützow, Mikko Loukovaara.

**Formal analysis:** Annukka Pasanen, Ralf Bützow, Mikko Loukovaara.

**Funding acquisition:** Ralf Bützow.

**Investigation:** Anne Kolehmainen, Annukka Pasanen, Taru Tuomi, Riitta Koivisto-Korander, Ralf Bützow, Mikko Loukovaara.

**Methodology:** Annukka Pasanen, Ralf Bützow, Mikko Loukovaara.

**Project administration:** Ralf Bützow, Mikko Loukovaara.

**Software:** Mikko Loukovaara.

**Supervision:** Riitta Koivisto-Korander, Ralf Bützow, Mikko Loukovaara.

**Validation:** Annukka Pasanen.

**Writing – original draft:** Anne Kolehmainen, Mikko Loukovaara.

**Writing – review & editing:** Anne Kolehmainen, Annukka Pasanen, Taru Tuomi, Riitta Koivisto-Korander, Ralf Bützow, Mikko Loukovaara.

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
