## [Decision Letter · Decision Letter 0]

28 Aug 2020

PONE-D-20-16843

Demographic factors as prognostic variables among molecular subgroups of endometrial cancer

PLOS ONE

Dear Dr. Loukovaara,

Thank you for submitting your manuscript to PLOS ONE. After careful consideration, we feel that it has merit but does not fully meet PLOS ONE’s publication criteria as it currently stands. Therefore, we invite you to submit a revised version of the manuscript that addresses the points raised during the review process.

Reviewers have provided very professional reviews, We look forward to receiving your revised manuscript.

Kind regards,

Ludmila Vodickova, M.D., PhD

Academic Editor

PLOS ONE

Journal Requirements:

2. Please provide additional details regarding participant consent. In the ethics statement in the Methods and online submission information, please ensure that you have specified (1) whether consent was informed and (2) what type you obtained (for instance, written or verbal). If your study included minors, state whether you obtained consent from parents or guardians. If the need for consent was waived by the ethics committee, please include this information.

Reviewers' comments:

Reviewer's Responses to Questions

**Comments to the Author**

1. Is the manuscript technically sound, and do the data support the conclusions?

Reviewer #1: Partly

Reviewer #2: Partly

2. Has the statistical analysis been performed appropriately and rigorously? 

Reviewer #1: Yes

Reviewer #2: Yes

3. Have the authors made all data underlying the findings in their manuscript fully available?

Reviewer #1: Yes

Reviewer #2: No

4. Is the manuscript presented in an intelligible fashion and written in standard English?

Reviewer #1: Yes

Reviewer #2: Yes

5. Review Comments to the Author

Reviewer #1: The authors conducted a single center retrospective study to evaluate the association between demographic factors such as age, body mass index and diabetes and outcomes among the different molecular subgroups of endometrial cancer. This is a topic of interest as our understanding of the underlying molecular biology of endometrial cancer has improved. Overall the manuscript is well written.

The abstract conclusion is too strong given the retrospective nature of the study and limited sample size within some of the subgroups. Recommend changing to "The prognostic effects of age, body mass index and type 2 diabetes do not appear to be uniform for the molecular subgroups of endometrial cancer." Also the data may support further evaluation of body mass index combined with genomics-based risk-assessment for discussion regarding prognosis but hard to argue implementation of BMI FOR genomic-based risk assessment.

Lines 164-166 - This section "Twenty cases (3.9%) displayed multiple molecular features. Three cases displayed POLE EDM and either MMR-D or p53 abn, and one case had all three molecular alterations. These were classified as POLE EDM tumors. Sixteen cases, classified as MMR-D tumors, displayed both MMR-D and p53 abn." belongs in the results section. In the methods please detail how you made this decision to categorize this cases in the manner you chose. Quit frankly this is problematic as these case may represent mixed histologic subtypes. While mixed endometrial cancers are rare they definitely are present and often the prognosis follows the highest grade. In the TCGA study 62% of the mixed histology clustered with CNH subgroup. While the authors used p53 protein expression as a surrogate for mutations this technology is not the same as some cancers with TP53 mutations may stain negative. Moreover, cancers could be grouped as CNH and not harbor a TP53 mutation (10%). Did histologic findings, ER/PR status, assist in distinguishing these cases further and what were the histologic and grade criteria for these 20 cases. If you rerun your analysis without these 20 cases do your results change?

Lines 312-313 The authors state "This may be the first study investigating associations of demographic factors with survival in the molecular subgroups of endometrial cancer described by the TCGA." you may want to consider deleting "This may be the first. . . " Also consider adding Dr. Roque manuscript "Association between differential gene expression and body mass index amon endometrial cancer from TCGA Project." Gynecology Oncol 2016 to support your findings.

Discussion - one issue to consider is that the patients evaluated in your study may not have a high frequency of morbid obesity. The worse survival outcomes are really seen in this group of patients. What is the percentage of patients with BMI>40 in each category? see Von Gruenigen et al paper regarding the outcomes of patients on GOG99.

Von Gruenigen VE, Tian C, Frasure H, Waggoner S, Keys H, Barakat RR. Treatment effects, disease recurrence, and survival in obese women with early endometrial carcinoma : a Gynecologic Oncology Group study. Cancer 2006;107(12):2786-91

There is also the issue of paradoxical obesity - obese patients tend to have a lower grade of disease and as you alluded may have a more favorable disease biology for those in the NSMP group.

Reviewer #2: This is an interesting study and as far as I know, it is the first to examine the impact of potentially important clinical predictors on endometrial cancer outcomes stratified by molecular subgroup.

The strengths of the manuscript include its novelty, large numbers from a single centre perspective, complete molecular subgroup assignment using standardised immunohistochemistry and POLE sequencing, long follow up and almost complete survival outcome data.

The weaknesses of the manuscript mostly relate to sample size since the study is underpowered to examine the associations between clinico-pathological factors and survival outcomes within each molecular subgroup, particularly when considering the POLE mutant and p53 subgroups. So really, this study can only examine the influence of clinical factors on outcomes from endometrial cancer in the NSMP and MMRd groups. Nonetheless, I think this manuscript is still of sufficient interest for publication since it will stimulate further research in this area and a sufficiently powered study would need several thousand patients - which is obviously unrealistic to achieve without a signal from a smaller study like this.

Considerations for the authors, to help improve the manuscript:

1. The English could be improved in places, for example, in the abstract "For POLE mutant (n = 37) and p53 aberrant (n = 69) subgroups, significant associations between demographic factors and overall mortality, cancer-related mortality and non-cancer related mortality were not observed during a median follow-up time of 81 months (range 1-136)." - this would definitely be better phrased ""For POLE mutant (n = 37) and p53 aberrant (n = 69) subgroups, no significant associations were observed between demographic factors and overall mortality, cancer-related mortality and non-cancer related mortality during a median follow-up time of 81 months (range 1-136)." This is a single example but there are multiple places where the English could do with a re-write.

2. The results would benefit from information regarding overall, cancer-specific and non-cancer deaths in the whole cohort and with respect to the 4 molecular subgroups. This isn't novel of course but it does help the reader to see that the expected outcomes are seen in this cohort and that the subgroups are behaving as predicted. If we know the cohort is representative, we can trust the subsequent analyses using the cohort. I think it needs a whole paragraph (and maybe Kaplan Meier curves for the 4 subgroups) detailing the outcomes of the whole cohort and then the outcomes by subgroup, including numbers of deaths. The death count is added as a footnote to a table at the moment, but that was the only place where it was visible to me.

3. The manuscript reads like a bit of a hotch potch rather than telling a story. I think it would benefit from a complete re-write, especially the introduction and discussion. There is a story here but the authors aren't telling it. The study is showing that tumour biology predominates in the excellent (POLE) and poor (p53) prognosis molecular subgroups, but in the intermediate prognosis groups, clinical factors become more important. It makes sense that cancer-related deaths are less common with obesity in the NSMP group where obesity predisposes to low grade, early stage endometrial cancer. It is not clear thus far the impact of obesity in the MMRd group, and this study helps to start documenting this. I think the introduction and discussion could be improved by telling the big picture first rather than individually reporting on the minutiae of each aspect of the study, without bringing it all together in a story.

Minor points for correction:

Line 25: Demographic factors have been suggested to be prognostic in endometrial cancer. Needs re-writing. What about "Clinical factors may influence endometrial cancer survival outcomes." BMI is not strictly demographic and the English as written is poor.

Line 28-30: Methods are incompletely described. No mention of years of accrual of cohort. No mention of setting. No mention of what survival outcomes, how they were obtained. No mention of stats used.

Line 31-33: See point 1

Line 43-45: Not keen on this conclusion. The last sentence should be deleted. Isn't your conclusion that tumour biology predominates for excellent (POLE) and poor prognostic (p53) subgroups, but in the NSMP and MMRd subgroups, clinical factors may influence survival outcomes in endometrial cancer?

Introduction: This is too long. It discusses the minutiae of the association of obesity, diabetes and age on endometrial cancer survival outcomes. It should be about one page in total length, give the big picture not the detail, explaining the rationale for the study set in the context of what the problem is you are trying to address.

Methods: What variables were included in the multivariable analysis. It is impossible to comment on the robustness of the analyses or the conclusions made without knowing what has been included here.

Results: Table 1 - please clarify how many patients had/ did not have adjuvant therapy before describing the detail of what they had. It appears that a very high proportion of patients had some form of adjuvant treatment even though there were lots of grade 1 stage 1a endometrial cancers in the cohort. This needs to be clarified.

Discussion: Much easier to follow if presented as a story. Start with a summary of the main findings before talking about your findings in the context of previous work, strengths/ limitations, clinical implications and future work. Limit to 4 pages.

6. PLOS authors have the option to publish the peer review history of their article (what does this mean?). If published, this will include your full peer review and any attached files.

Reviewer #1: No

Reviewer #2: No

---

## [Author Response · Author response to Decision Letter 0]

22 Sep 2020

Ludmila Vodickova, MD, PhD

Academic Editor

PLOS ONE

September 20, 2020

Dear Editor:

Thank you for your e-mail of August 28. We have addressed the questions raised by the Journal and Reviewers as follows:

Additional Journal Requirements:

Author response: PLOS ONE style requirements are met.

2. Please provide additional details regarding participant consent. In the ethics statement in the Methods and online submission information, please ensure that you have specified (1) whether consent was informed and (2) what type you obtained (for instance, written or verbal). If your study included minors, state whether you obtained consent from parents or guardians. If the need for consent was waived by the ethics committee, please include this information.

Author response: Participant consent was waived because this was a retrospective study where consent was difficult or impossible to obtain. This is now stated in the manuscript (lines 108-109). Instead of participant consent, the institutional review board called for an approval by the National Supervisory Authority for Welfare and Health, which was granted (lines 107-108).

Reviewer #1:

The abstract conclusion is too strong given the retrospective nature of the study and limited sample size within some of the subgroups. Recommend changing to "The prognostic effects of age, body mass index and type 2 diabetes do not appear to be uniform for the molecular subgroups of endometrial cancer." Also the data may support further evaluation of body mass index combined with genomics-based risk-assessment for discussion regarding prognosis but hard to argue implementation of BMI FOR genomic-based risk assessment.

Author response: The abstract conclusion has been changed as suggested by the Reviewer.

Lines 164-166 - This section "Twenty cases (3.9%) displayed multiple molecular features. Three cases displayed POLE EDM and either MMR-D or p53 abn, and one case had all three molecular alterations. These were classified as POLE EDM tumors. Sixteen cases, classified as MMR-D tumors, displayed both MMR-D and p53 abn." belongs in the results section. In the methods please detail how you made this decision to categorize this cases in the manner you chose. Quit frankly this is problematic as these case may represent mixed histologic subtypes. While mixed endometrial cancers are rare they definitely are present and often the prognosis follows the highest grade. In the TCGA study 62% of the mixed histology clustered with CNH subgroup. While the authors used p53 protein expression as a surrogate for mutations this technology is not the same as some cancers with TP53 mutations may stain negative. Moreover, cancers could be grouped as CNH and not harbor a TP53 mutation (10%). Did histologic findings, ER/PR status, assist in distinguishing these cases further and what were the histologic and grade criteria for these 20 cases. If you rerun your analysis without these 20 cases do your results change?

Author response: Data on multiple classifiers can now be found in the results section (lines 182-186). Categorization of multiple classifiers was based on clinical outcomes of these tumors, as reported in recent literature (lines 125-126, references #11 and #12). Given the fact that the TCGA classification system is not primarily based on morphology, our approach may be more relevant than categorization according to histologic criteria (or ER/PR status). Multiple classifiers were categorized either as POLE EDM (n = 4) or MMR-D (n = 16). Their histology distributions were as follows:

-POLE EDM (n = 4): 1 grade 2 endometrioid carcinoma, 1 grade 3 endometrioid carcinoma, 1 clear cell carcinoma, and 1 serous carcinoma

-MMR-D (n = 16): 6 grade 2 endometrioid carcinomas, 7 grade 3 endometrioid carcinomas, 1 serous carcinoma, 1 carcinosarcoma, and 1 undifferentiated carcinoma

These data are not shown in the manuscript because the number of multiple classifiers was too small for a meaningful comparison with single classifiers.

Hazard ratios for mortality outcomes remained essentially unaltered after exclusion of multiple classifiers from the POLE EDM and MMR-D subgroups (not shown).

Lines 312-313 The authors state "This may be the first study investigating associations of demographic factors with survival in the molecular subgroups of endometrial cancer described by the TCGA." you may want to consider deleting "This may be the first. . . " Also consider adding Dr. Roque manuscript "Association between differential gene expression and body mass index among endometrial cancer from TCGA Project." Gynecology Oncol 2016 to support your findings.

Author response: Roque et al. evaluated differences in gene expression profiles of obese and non-obese women with endometrial cancer and examined the association of BMI within the clusters identified in TCGA. However, associations of BMI with survival outcomes were not examined. Regardless, we have rephrased the sentence in the discussion as suggested (lines 310-311). The article by Roque et al. is now cited (reference #27).

Discussion - one issue to consider is that the patients evaluated in your study may not have a high frequency of morbid obesity. The worse survival outcomes are really seen in this group of patients. What is the percentage of patients with BMI>40 in each category? see Von Gruenigen et al paper regarding the outcomes of patients on GOG99.

Author response: The proportion of morbidly obese patients (BMI ≥40 kg/m2) in each molecular subgroup can now be found in Table 1. The rate of morbid obesity was too small for further analyses.

There is also the issue of paradoxical obesity - obese patients tend to have a lower grade of disease and as you alluded may have a more favorable disease biology for those in the NSMP group.

Author response: The multivariable analysis of cancer-related mortality in the NSMP subgroup included histology and grade as confounders (Table 4). The hazard ratio for overweight/obesity remained significant in this analysis. It could therefore be assumed that BMI has an independent effect on patient outcome in this subgroup.

Reviewer #2:

Considerations for the authors, to help improve the manuscript:

1. The English could be improved in places, for example, in the abstract "For POLE mutant (n = 37) and p53 aberrant (n = 69) subgroups, significant associations between demographic factors and overall mortality, cancer-related mortality and non-cancer related mortality were not observed during a median follow-up time of 81 months (range 1-136)." - this would definitely be better phrased ""For POLE mutant (n = 37) and p53 aberrant (n = 69) subgroups, no significant associations were observed between demographic factors and overall mortality, cancer-related mortality and non-cancer related mortality during a median follow-up time of 81 months (range 1-136)." This is a single example but there are multiple places where the English could do with a re-write.

Author response: This sentence in the abstract was deleted as the abstract was for the most part rewritten. The whole manuscript has now been edited by a professional language editing service (Helsinki University Language Center).

2. The results would benefit from information regarding overall, cancer-specific and non-cancer deaths in the whole cohort and with respect to the 4 molecular subgroups. This isn't novel of course but it does help the reader to see that the expected outcomes are seen in this cohort and that the subgroups are behaving as predicted. If we know the cohort is representative, we can trust the subsequent analyses using the cohort. I think it needs a whole paragraph (and maybe Kaplan Meier curves for the 4 subgroups) detailing the outcomes of the whole cohort and then the outcomes by subgroup, including numbers of deaths. The death count is added as a footnote to a table at the moment, but that was the only place where it was visible to me.

Author response: A column showing mortality data for the whole cohort has been added to Table 2. Numbers of deaths in the whole cohort and in each subgroup are shown as a footnote in Table 2. Numbers of deaths in the whole cohort are now also reported in the text (lines 192-193).

3. The manuscript reads like a bit of a hotch potch rather than telling a story. I think it would benefit from a complete re-write, especially the introduction and discussion. There is a story here but the authors aren't telling it. The study is showing that tumour biology predominates in the excellent (POLE) and poor (p53) prognosis molecular subgroups, but in the intermediate prognosis groups, clinical factors become more important. It makes sense that cancer-related deaths are less common with obesity in the NSMP group where obesity predisposes to low grade, early stage endometrial cancer. It is not clear thus far the impact of obesity in the MMRd group, and this study helps to start documenting this. I think the introduction and discussion could be improved by telling the big picture first rather than individually reporting on the minutiae of each aspect of the study, without bringing it all together in a story.

Author response: The introduction and discussion have been rewritten. The introduction is now shorter. The discussion begins with a summary of the main findings.

Minor points for correction:

Line 25: Demographic factors have been suggested to be prognostic in endometrial cancer. Needs re-writing. What about "Clinical factors may influence endometrial cancer survival outcomes." BMI is not strictly demographic and the English as written is poor.

Author response: Corrected as suggested by the Reviewer (line 24). The same correction (word “clinical” instead of “demographic”) has been made throughout the manuscript, including the title.

Line 28-30: Methods are incompletely described. No mention of years of accrual of cohort. No mention of setting. No mention of what survival outcomes, how they were obtained. No mention of stats used.

Author response: The abstract methods has been complemented as suggested by the Reviewer.

Line 31-33: See point 1

Author response: Please see our response to point 1.

Line 43-45: Not keen on this conclusion. The last sentence should be deleted. Isn't your conclusion that tumour biology predominates for excellent (POLE) and poor prognostic (p53) subgroups, but in the NSMP and MMRd subgroups, clinical factors may influence survival outcomes in endometrial cancer?

Author response: The abstract conclusion has been changed as suggested by Reviewer #1, please see above.

Introduction: This is too long. It discusses the minutiae of the association of obesity, diabetes and age on endometrial cancer survival outcomes. It should be about one page in total length, give the big picture not the detail, explaining the rationale for the study set in the context of what the problem is you are trying to address.

Author response: Please see our response to point 3.

Methods: What variables were included in the multivariable analysis. It is impossible to comment on the robustness of the analyses or the conclusions made without knowing what has been included here.

Author response: Variables included in the multivariable analysis are found in Table 4. They can now also be found in the Methods section, along with appropriate references (lines 173-175).

Results: Table 1 - please clarify how many patients had/ did not have adjuvant therapy before describing the detail of what they had. It appears that a very high proportion of patients had some form of adjuvant treatment even though there were lots of grade 1 stage 1a endometrial cancers in the cohort. This needs to be clarified.

Author response: A new line has been added to Table 1, showing the number of patients who did not receive any adjuvant therapies. Stratification of patients to adjuvant therapies is outlined in Materials and methods (lines 86-92).

Discussion: Much easier to follow if presented as a story. Start with a summary of the main findings before talking about your findings in the context of previous work, strengths/ limitations, clinical implications and future work. Limit to 4 pages.

Author response: Please see our response to point 3.

We thank the Reviewers for the valuable comments which helped us to improve the presentation considerably. A clean and marked version of the revised manuscript have been uploaded in the editorial system. In the marked version, changes are shown as red and strikethrough text. We hope that the revised manuscript will be found suitable for publication in PLOS ONE.

With kind regards,

Mikko Loukovaara, MD

Department of Obstetrics and Gynecology

Helsinki University Hospital and University of Helsinki

P.O. Box 140, 00029 HUS, Helsinki, Finland

Tel.: +358504272526

Fax: +358947173640

E-mail: mikko.loukovaara@hus.fi

---

## [Decision Letter · Decision Letter 1]

9 Nov 2020

Clinical factors as prognostic variables among molecular subgroups of endometrial cancer

PONE-D-20-16843R1

Dear Dr. Loukovaara,

We’re pleased to inform you that your manuscript has been judged scientifically suitable for publication and will be formally accepted for publication once it meets all outstanding technical requirements.

Kind regards,

Ludmila Vodickova, M.D., PhD

Academic Editor

PLOS ONE

Reviewers' comments:

Reviewer's Responses to Questions

**Comments to the Author**

1. If the authors have adequately addressed your comments raised in a previous round of review and you feel that this manuscript is now acceptable for publication, you may indicate that here to bypass the “Comments to the Author” section, enter your conflict of interest statement in the “Confidential to Editor” section, and submit your "Accept" recommendation.

Reviewer #2: All comments have been addressed

2. Is the manuscript technically sound, and do the data support the conclusions?

Reviewer #2: Yes

3. Has the statistical analysis been performed appropriately and rigorously? 

Reviewer #2: I Don't Know

4. Have the authors made all data underlying the findings in their manuscript fully available?

Reviewer #2: Yes

5. Is the manuscript presented in an intelligible fashion and written in standard English?

Reviewer #2: Yes

6. Review Comments to the Author

Reviewer #2: The manuscript is much improved. Please do not use the term 'morbid obesity'. It is pejorative and labelling. Please instead refer simply to BMI>40 or, if you must classify, to WHO class III obesity. Since the term is mentioned only twice in the whole manuscript, with one of those times to define it, my guess would be you can stick to BMI >40 and refer to overweight/obesity as BMI>25.

7. PLOS authors have the option to publish the peer review history of their article (what does this mean?). If published, this will include your full peer review and any attached files.

Reviewer #2: No

---

## [Editor Report · Acceptance letter]

13 Nov 2020

PONE-D-20-16843R1 

Clinical factors as prognostic variables among molecular subgroups of endometrial cancer 

Dear Dr. Loukovaara:

I'm pleased to inform you that your manuscript has been deemed suitable for publication in PLOS ONE. Congratulations! Your manuscript is now with our production department. 

Kind regards, 

on behalf of

Dr. Ludmila Vodickova 

Academic Editor

PLOS ONE